# Novel Fluorinated Poly (Lactic-Co-Glycolic acid) (PLGA) and Polyethylene Glycol (PEG) Nanoparticles for Monitoring and Imaging in Osteoarthritis

**DOI:** 10.3390/pharmaceutics13020235

**Published:** 2021-02-07

**Authors:** Luana Zerrillo, Karthick Babu Sai Sankar Gupta, Fons A.W.M. Lefeber, Candido G. Da Silva, Federica Galli, Alan Chan, Andor Veltien, Weiqiang Dou, Roberta Censi, Piera Di Martino, Mangala Srinivas, Luis Cruz

**Affiliations:** 1Translational Nanobiomaterials and Imaging (TNI) Group, Department of Radiology, Leiden University Medical Centrum, 2333 ZA Leiden, The Netherlands; 2Percuros-Tarweveld 5, 6617 CD Bergharen, The Netherlands; 3Institute of Chemistry, Leiden University, P.O. Box 9502, 2300 RA Leiden, The Netherlands; 4Leiden Institute of Physics, Niels Bohrweg 2, 2333 CA Leiden, the Netherlands; 5Department of Tumor Immunology, Radboud Institute for Molecular Life Sciences (RIMLS), Geert Grooteplein Zuid 28, 6525 GA Nijmegen, The Netherlands; 6School of Pharmacy, University of Camerino, 62032 Camerino, Italy; 7Department of Tumor Immunology, Radboud University Medical Center, 6525 XZ Nijmegen, The Netherlands

**Keywords:** PLGA nanoparticles, (-) (2-trifluoroacetamide) succinic anhydride (TFA), drug delivery, intra-articular injection, molecular imaging, fluorine-based magnetic resonance imaging (^19^F-MRI), osteoarthritis

## Abstract

Polymeric nanoparticles (NPs) find many uses in nanomedicine, from drug delivery to imaging. In this regard, poly (lactic-co-glycolic acid) (PLGA) and polyethylene glycol (PEG) particles are the most widely applied types of nano-systems due to their biocompatibility and biodegradability. Here we developed novel fluorinated polymeric NPs as vectors for multi-modal nanoprobes. This approach involved modifying polymeric NPs with trifluoroacetamide (TFA) and loading them with a near-infrared (NIR) dye for different imaging modalities, such as magnetic resonance imaging (MRI) and optical imaging. The PLGA-PEG-TFA NPs generated were characterized in vitro using the C28/I2 human chondrocyte cell line and in vivo in a mouse model of osteoarthritis (OA). The NPs were well absorbed, as confirmed by confocal microscopy, and were non-toxic to cells. To test the NPs as a drug delivery system for contrast agents of OA, the nanomaterial was administered via the intra-articular (IA) administration method. The dye-loaded NPs were injected in the knee joint and then visualized and tracked in vivo by fluorine-19 nuclear magnetic resonance and fluorescence imaging. Here, we describe the development of novel intrinsically fluorinated polymeric NPs modality that can be used in various molecular imaging techniques to visualize and track OA treatments and their potential use in clinical trials.

## 1. Introduction

The causes of osteoarthritis (OA) are not fully understood. Almost 10% of the population suffers from symptomatic knee OA at the age of 60, a condition that leads to articular cartilage damage [1,2]. The cartilage is the main component of synovial joints and it is formed by chondrocytes. These cells assemble the extracellular matrix and they contribute greatly to hyaline cartilage regeneration. In knee joint OA, physiological changes are observed in the synovial membrane, cartilage, and bone structures. There are no definite pharmacological treatments for OA. This disease is divided into stages. Although there are no pharmacological treatments for the early stage, OA progression can be slowed by lifestyle changes such as weight loss and physical exercise. As the disease progresses, pharmacological treatments such as anti-inflammatory drugs, and intra-articular (IA) injections of corticosteroids or hyaluronic acid can be administered. In the final stage of OA, there are no drugs that can relieve the pain, and surgery is the only therapeutic option available [3,4].

Since 1986, knee OA has been diagnosed through the combination of radiographs and physical examination. Recently, magnetic resonance imaging (MRI) has received considerable attention because of its capacity to provide high-resolution deep tissue imaging. This technique allows visualization of all tissues in the joint, including cartilage, menisci, bone, and soft tissue [5,6]. Moreover, ^19^F-MRI has the advantage over H-MRI in that it reduces background signals. ^19^F molecules provide excellent signal sensitivity and high specificity compared to ^1^H because of the lack of background signal from endogenous fluorine [7]. In this context, perfluorocarbon (PFC)-labeled probes can facilitate ^19^F-MRI imaging of particles, hydrogels, and drugs in the body [8]. Furthermore, some PFCs have also been approved for medical applications, such as contrast-enhanced ultrasound [7]. Overall, PFCs are known to be well-tolerated in vivo. They are non-toxic molecules and normally diffuse back into the blood, where they ultimately dissolve into plasma lipids. A typical ^19^F-MRI probe contains fluorine compounds like perfluoro decalin, perfluoro hexane, perfluorooctyl bromide, perfluoro-15-crown 5-ether, and trifluoroacetamide (TFA) [7,9]. Here we used TFA as the PFC probe as it can be covalently coupled to polymers, biomolecules, and small drugs and can be used for visualization and tracking by ^19^F-MRI. Therefore, in our approach, we coupled TFA to poly (lactic-co-glycolic acid (PLGA) and polyethylene glycol (PEG)-based nanoparticles (NPs). Among the polymers developed to formulate polymeric NPs, PLGA has attracted considerable attention due to the following [10]: (i) It shows biodegradability and biocompatible properties. PLGA is hydrolyzed into its monomeric components, lactic and glycolic acid, which are natural metabolites of the human body; (ii) PLGA-based drugs products have been approved by the FDA and European Medicine Agency for parenteral administration as drug delivery systems; (iii) it has well-described formulations and methods of production that can be adapted to various types of drugs and contrast agents, such as hydrophilic or hydrophobic small molecules and macromolecules; (iv) it can protect the drug and contrast agent from degradation; (v) it can sustain drug release; and (vi) the surface properties can be modified to provide “stealth” and/or better interaction with biological materials. PLGA can conjugate many targeting and imaging moieties on a single particle to enhance binding affinity and specificity, and it provides amplified signals at the target region. Furthermore, it can be used to engineer NPs for effective and targeted delivery of imaging labels, prolonged plasma half-lives, enhancing stability, improving targeting efficiency, and reducing non-specific binding; and (vii) it can be modified to target NPs to specific organs or cells [11,12]. Conjugation and loading of other contrast agents, such as PFC or NIR dye, is an obvious advantage of PLGA because this polymer prevents unwanted dispersion during trafficking to the OA site. PEG is a hydrophilic, non-ionic polymer with high biocompatibility that can be added to NPs via various approaches, including covalent binding. The presence of PEG on the surface of PLGA NPs confers other advantages, such as increased half-life in systemic circulation and muco-adhesion of the particles [11,12,13]. Given these observations, we chose PLGA-PEG NPs containing conjugated fluorine to track and visualize the NPs in vivo in osteoarthritic knee mice using ^19^F-MRI.

Here, we synthesized and developed a new copolymer formed by PLGA-PEG-TFA for multimodal proposes. The NPs were successfully traced by optical imaging and ^19^F-MRI using NIR dye loaded into the particles and TFA present on their shell. We studied the capacity of the novel NPs for drug delivery of contrast agents in OA. The NPs were characterized physically and chemically and then tested in vitro in C28/I2 human chondrocytes. Finally, they were also injected intraarticularly into a mouse model of OA and traced in vivo by optical imaging and ^19^F-MRI.

## 2. Materials and Methods

### 2.1. Materials

PLGA (PURASORB^®^ PDLG 5002A 50:50, with an inherent viscosity of 0.20 dL/g, MW = 17,000) was obtained from Carbion PURAC (Amsterdam, The Netherlands). Polyvinyl alcohol (PVA) (87–89% hydrolyzed, typical MW 13.000–23.000), *N*-(3-dimethylaminopropyl)-*N*′-ethyl carbodiimide hydrochloride (EDC), *N*-hydroxysuccinimide (NHS), methylene chloride, dimethylformamide, and chloroform, (-)(2-trifluoroacetamide) succinic anhydride and triethylamine (TEA) were purchased from Sigma-Aldrich (Steinem, Germany). PEG with diamine group (NH_2_-PEG-NH_2_, MW = 220.31) was supplied by Shanghai Seebio Biotech Inc. (Shanghai, People’s Republic of China). The solvent dichloromethane (DCM), used to prepare the NPs, was purchased from Sigma Aldrich (Steinem, Germany). The NPs were all loaded with NIR dye (IR-780 Iodide) purchased from Sigma-Aldrich (Zwijndrecht, The Netherlands). The in vitro studies were performed using the CellTiter 96^®^ AQueous One Solution Cell Proliferation Assay (Promega, Walldorf, Germany), Dulbecco’s Modified Eagles Medium (Thermo Fisher Scientific, Waltham, MA, USA), and fetal calf serum (FCS; Life Technologies, Eugene, OA, USA). To-pro 3 iodide (642/661) was purchased from Invitrogen (Thermo Fisher Scientific, Waltham, MA, USA); [3-(4,5-dimethylthiazol-2-yl)-5-(3-carboxymethoxyphenyl)-2-(4-sulfophenyl)-2H-tetrazolium (MTS) was supplied by Promega (Leiden, The Netherlands).

### 2.2. Copolymer Formulation

PLGA-PEG-(2-trifluoroacetamide) succinic anhydride (TFA) was synthesized by the conjugation of NH_2_–PEG–NH_2_ with PLGA–COOH. PLGA–COOH (5 g, 0.28 mmol). It was then dissolved in methylene chloride (15 mL) and converted to PLGA–NHS with excess *N*-hydroxy succinimide (NHS, 260 mg, 2.3 mmol) in the presence of 1-ethyl-3-(3-dimethylaminopropyl)-carbodiimide (EDC, 222 mg, 2.3 mmol). PLGA–NHS was precipitated with cold diethyl ether (15 mL) and washed three times in an ice-cold mixture of 80% ethyl ether and 20% methanol to remove residues of NHS. After drying under vacuum, PLGA–NHS (3 g, 0.173 mmol) was dissolved in chloroform (5 mL) and with NH_2_–PEG–NH_2_ (760 mg, 0.3 mmol) and *N*, *N*-diisopropylethylamine (22 mg, 0.173 mmol). The copolymer was precipitated with cold methanol for 24 h and washed three times in an ice-cold mixture of 80% ethyl ether and 20% methanol to remove unreacted PEG. The resulting PLGA–PEG-NH_2_ block copolymer was dialyzed in water and lyophilized. Subsequently, the compounds were characterized by ^1^H-NMR.

After lyophilization, PLGA-PEG-NH_2_ (0.40 g, 0.22 mmol) was dissolved in dimethylformamide (DMF) (10 mL). When the copolymers were completely dissolved, (-) (2-trifluoroacetamide) succinic anhydride (0.01 g, 0.44 mmol) and triethylamine (TEA) (0.22 mmol) were added to the solution and the reaction was performed for 24 h at room temperature. The copolymers PLGA-PEG-TFA were dialyzed in water and lyophilized. Later, the compounds were characterized by ^19^F-NMR.

### 2.3. Preparation of NPs

#### Preparation of PLGA-PEG and PLGA-PEG-TFA NPs

PLGA-PEG and PLGA-PEG-TFA NPs were generated and loaded with NIR dye (IR-780 Iodide). All NPs were prepared by the single emulsion-solvent evaporation method to reduce the particle size to the submicron range. NPs were dissolved in DCM together with NIR dye and emulsified in water phase containing a PVA (0.5% (*w*/*v*)).

The emulsions were continuously stirred at 24,000 rpm using a mechanical stirring system (ULTRA-TURRAX T25, IKA, Staufen, Germany) until the organic solvent partitioned into the aqueous phase and then evaporated. The emulsion formed was stirred under magnetic stirring (IKA^®^ RCT basic IKAMAG™, Staufen, Germany) for >20 min to evaporate the solvent. To complete solvent evaporation, the NP suspensions were stirred overnight with a magnetic stirrer. The suspensions formed were separated from the excess of PVA by ultracentrifugation at 12,000 rpm and 4 °C for 30 min. The NPs were then washed three times with MilliQ water to remove any potential PVA remaining. Finally, the pellets consisting of NPs were re-suspended in MilliQ water and lyophilized.

To prepare the PLGA-PEG and PLGA-PEG-TFA NPs loaded with fluorescein isothiocyanate (FITC) dye, a similar protocol as the one described above for the NIR dye was used, with the exception that 0.5 mg of FITC dye was initially dissolved in MilliQ water.

### 2.4. Characterization of NPs

#### 2.4.1. Morphology and Size Characterization

The morphology of the two types of NP was studied by atomic force microscopy (AFM, JPK NanoWizard 3 AFM, by JPK Instruments (Bruker), Berlin, Germany.). The NPs were diluted and dispersed in MilliQ water. A drop of the suspension was placed on a clean glass surface glued to the AFM stub and air-dried for 1 h. The dried NPs were imaged by AFM (JPK Nano Wizard 3) in AC mode (tapping mode) using OMCL-AC160TS silicon probes from Olympus with a nominal resonance frequency of 300 kHz and a nominal spring constant of 26 N/m. The images were analyzed using Gwyddion SPM Software (Version 2.5, Open Source Software, http://gwyddion.net/)).

The average size and polydispersity index (PDI) of the NPs were determined by Dynamic Light Scattering (DLS) (Zetasizer Nano S90, Malvern Instruments, Worcestershire, Cambridge, UK). NPs were dispersed in MilliQ water and the measurements were performed at 25 °C and an angle of 90°. Averages and standard deviations were calculated from triplicate measurements. The stability or the aggregation of NPs was determined by Zeta potential (Zetasizer Nano S90, Malvern Instruments, Worcestershire, UK).

#### 2.4.2. Encapsulation Efficiency Analysis of NIR Dye

To determine the encapsulation efficiency (EE) and the loading content of NIR dye, the lyophilized NPs were dissolved in 0.8 M NaOH. Next, 5 mg of PLGA-PEG and PLGA-PEG-TFA NPs was dissolved separately in 0.5 mL 0.8 M NaOH overnight at 37 °C. Afterwards, the basic solutions of all the NPs were centrifuged at 12,000 rpm at room temperature for 20 min and the resultant supernatants were collected. The NIR dye content was then measured using an Odyssey Infrared Imager 9120 (LI-COR) scanner at 800 nm. The EE for NIR dye was calculated using the formula [14] below:$$EE=\frac{Amount\;of\;drug\;in\;formulation}{Amount\;of\;drug\;used\;for\;formulation}\times 100.$$

*Amount of drug in formulation* is the amount of NIR dye loaded in the NPs. The values were obtained as described above. *Amount of drug used for formulation* is the amount of NIR dye added to the preparation of NPs.

#### 2.4.3. Encapsulation Efficiency Analysis of FITC dye

To study the EE for FITC dye, 5 mg of dry NPs was dissolved in 0.8 M NaOH, as described above. The separate solutions containing PLGA-PEG and PLGA-PEG-TFA NPs were centrifugated at 12,000 rpm for 20 min at room temperature, and the supernatants were collected. The FITC dye content was then measured using an Amersham Biosciences Ultrospec 2100 pro, UV/Vis Spectrophotometer. The amount of FITC dye was calculated from the linear regression of the standard curve obtained from the FITC solution.

#### 2.4.4. Study of NIR Dye in PBS Released from NPs

The release study of NIR dye was performed for 30 days, as previously described [15]. Briefly, 5 mg of lyophilized PLGA-PEG and PLGA-PEG-TFA NPs were dissolved in 5 mL of saline phosphate buffer (PBS). The NP solutions were gently stirred at 25/27 °C. At 0, 3, 9, 12, 14, 17, 23, 27, and 30 days, the solutions were centrifuged for 20 min at 12,000 rpm, and 150 μL of the supernatant was collected and replaced with 150 μL of fresh PBS. The NIR dye of each aliquot was then measured using an Odyssey™ scanner at 800 nm.

#### 2.4.5. Relaxivity Measurements

To assess the relaxation characteristics of the PLGA-PEG-TFA NPs as a T1-T2 dual MR contrast agent, their T1 and T2 relaxation times were measured with a 3T MRI (SIEMENS MAGNETOM Trio I-class, Erlangen, Germany). All samples were diluted in D_2_O. Measurements were performed using av500 instrument which is 11.7 tesla.

### 2.5. In Vitro Experiments

#### 2.5.1. Cell Culture

C28/I2 human chondrocytes are used as model cells for studying normal and pathological cartilage repair [16]. Cells were cultured in 75-cm flasks in 1:1 Dulbecco’s Modified Eagles Medium (DMEM)/F12 medium (Gibco Cell Culture Medium, ThermoFisher Scientific Magdeburg, Germany) with 10% (*v*/*v*) FCS at 37 °C and an atmosphere of 95% air and 5% CO_2_. The medium was replaced every 48 h. The cells were sub-cultured for experiments after reaching approximately 80–90% confluence [17].

#### 2.5.2. Cell Metabolic Assay (MTS)

To test the cell metabolism of the NPs, 5 × 10^4^ C28/I2 cells/well were seeded in 96-well plates. After 24 h of incubation at 37 °C, the NPs were added to the cells at concentrations of 10, 20, 40, 60, and 80 µg/mL. The MTS assay was performed at 24, 48, and 72 h. To establish a positive cytotoxicity control, cells were treated with 50% DMSO. Non-treated cells were used as a negative control. At 24, 48, and 72 h, 20 µL of [3-(4,5-dimethylthiazol-2-yl)-5-(3-carboxymethoxyphenyl)-2-(4-sulfophenyl)-2H-tetrazolium (MTS) was added and incubated at 37 °C for 1 h. The absorbance was measured with a spectrophotometer at λex 590 nm (Molecular Devices VERSAmax Tunable Microplate Reader, LUMC, Leiden, The Netherlands). The assay was assessed following the manufacturer’s instructions. Cellular metabolic activity in each condition is expressed as percentage increase in relation to untreated controls.

#### 2.5.3. NP Uptake Study by Optical Imaging

The C28/I2 cells were seeded in 96-well cell culture microplates (Greiner Bio-One B.V., Alphen aan den Rijn, The Netherlands) (2.5 × 10^4^ cells/well) and incubated with 40 µg/mL of PLGA-PEG or PLGA-PEG-TFA NPs for 1, 2, 4, 8, 24, 48, and 72 h. After incubation, the cells were washed twice with PBS, fixed for 15 min with 2% paraformaldehyde in PBS, rinsed in PBS, and stained with TO-PRO^®^-3 iodide dye (Molecular Probes, ThermoFisher, Marietta, OH, USA), which makes the cell nucleus detectable at 700 nm. Plates were analyzed using an Odyssey Infrared Imager 9120 (LI-COR, Lincoln, NE, USA) scanner at 800 nm and 700 nm for visualization of NPs loaded with NIR and the cells, respectively.

#### 2.5.4. NP Uptake Study by Confocal Microscopy

To examine the cellular uptake of NPs, 1 × 10^3^ cells/well were seeded in glass-bottom microwell dishes (MatTek Corporation) for 12 h and then incubated with PLGA-PEG or PLGA-PEG-TFA NPs at 37 °C for 24 h. The cells were then washed twice with PBS and fixed with 4% paraformaldehyde for 10 min. Cell membranes were then stained with Did (1:400) and the nuclei with DAPI (Invitrogen by Thermo Fisher Scientific). The cells and NPs were imaged using a Leica SP5C Spectral Confocal Laser Scanning Microscope. The NPs loaded FITC dye was used in this experiment instead of NIR dye.

#### 2.5.5. NP Uptake Studies by Fluorescence Microscopy

To visualize the uptake of NPs by cells after 24 h of incubation, an additional experiment was performed. The cells were seeded in a chamber plate (Falcon™ Chambered Cell Culture Slides) (2.5 × 104 cells/well) containing coverslips and co-incubated for 24 h with 40 μg/mL of PLGA-PEG or PLGA-PEG-TFA NPs. After incubation, the cells were washed twice with PBS, fixed for 15min with 2% paraformaldehyde in PBS and rinsed in PBS. Cell membranes were stained with CD44-PE (CD44 Monoclonal Antibody (IM7), PE, and eBioscience™) for 20 min at 37 °C and then covered with Vectashield mounting medium containing DAPI. Coverslips were mounted using Acqua Poly/Mount (Polysciences) and examined by fluorescent microscopy (Leica DMRA fluorescence microscopy, LUMC, University of Leiden).

### 2.6. In Vivo and Ex Vivo NP Characterization

#### 2.6.1. Optical Imaging

The animal procedures were conducted at the Leiden University Medical Center and approved by the Animal Welfare Committee (approval number 12036). The retention time of fluorescent PLGA-PEG-TFA NPs loaded with NIR dye was studied by optical imaging. To this end, 12-week-old C57BL/6 Jico mice were purchased from Charles River, France. The mice were divided into three groups (*n* = 3); their right knees were subjected to surgical destabilization of medial meniscus (DMM). Another three mice were used as controls. After 3 weeks of OA induction, the mice received an intra-articular (IA) injection of 8 µL of the NPs dissolved in water at a concentration of 20 mg/mL. To confirm the presence of fluorescent NPs in the knee, the mice were anesthetized using isoflurane and scanned using a Pearl^®^ Trilogy Small Animal Imaging System at 800 nm. After confirmation, the animals were sacrificed at different time points: Group 1 at 2 h post-injection; group 2 at 48 h and group 3 at 166 h. For each group, a control mouse was also sacrificed. At the end of the experiment, the knees and organs of the animals were collected and used for ex vivo experiments. The organs (heart, liver, spleen, lungs, and kidneys) were collected, weighed and scanned at 800 nm in an Odyssey Imaging System. The bio-distribution data for the organs are expressed as a percentage Injected Dose per gram of tissue (%ID/g).

#### 2.6.2. ^19^F-MRI

All samples were scanned in an 11.7 T MRI system (BioSpec, Bruker, Germany) equipped with a 1H/19F volume coil. 1H/19F MR images of whole mouse legs (fixed in PFA) were acquired by a zero echo time (ZTE) sequence with the following parameters: Repetition time (TR) = 2 ms/2 ms, image resolution = 0.23 × 0.23 × 0.23/0.47 × 0.47 × 0.47 mm, 1 average/64 averages, flip angle (FA) = 2°/4°, and acquisition time (TA) = 0:06:53/1:50:43 h. The 1H and 19F MR images obtained were overlaid using MRIcro software (Smith Micro software, Aliso Vijeo, CA, USA).

#### 2.6.3. µ-. CT Scan

A SkyScan 1076 µ-CT scanner (Bruker, Kontich, Belgium) was used after sacrificing the mice. Hind limbs were fixed in formalin and scanned at a voltage of 40 kV and current of 250 uA, with an X-ray source rotation step size of 0.8° over 180°. Images were taken with an image pixel resolution of 9.03 μm and an average frame of 4 to reduce noise. Reconstructions were made using the nRecon V1.6.2.0 software (Bruker) with a beam hardening correction set to 25%, a ring artifact correction set to 5, and the dynamic range set to −1000–4000 Hounsfield units. 3D imaging was performed with Cyttron Visualization Platform software (LUMC, Leiden, The Netherlands).

#### 2.6.4. Histological Analysis

After optical imaging and ^19^F-MRI examinations, the limbs were fixed with 4% phosphate-buffered paraformaldehyde for 24 h, decalcified with 10% ethylenediaminetetraacetic acid (pH 7.4) for 2 weeks, and then embedded in paraffin. The knee joints were sliced into 5 μm sections, stained with Hematoxylin/Eosin, F4/80 and Safranin O/Fast green examined by light microscopy to evaluate macrophages presence and the cartilage damage of the femur and tibia in the knee joint.

### 2.7. Statistical Analysis

Except for the in vivo experiment, all experiments were performed in triplicate. Graph Pad Prism software version 5 (La Jolla, CA, USA) was used for statistical analysis. Data were analyzed by Students’ *t*-test and two-way analysis of variance (ANOVA). In all analyses, significant difference was inferred at α = 0.05.

## 3. Results and Discussion

### 3.1. Synthesis and Characterization of the PLGA–PEG–TFA Copolymer

Our novel PLGA-PEG-TFA copolymer was synthesized in three steps. First, the carboxyl-group of PLGA was activated with excess NHS/EDC. Next the activated carboxyl group of PLGA was directly conjugated with excess NH_2_–PEG–NH_2_ [18]. The third and final step was the conjugation of PLGA-PEG-NH_2_ to TFA (Figure 1). After preparing the copolymer, the efficiency of the block copolymer reaction was determined by ^1^H-NMR (Figure 2A). To confirm the binding of TFA to the copolymers, ^19^F-NMR was measured (Figure 2B). The ^1^H-NMR spectrum of PLGA-PEG-TFA showed the typical peak at 3.5 ppm; from the methylene groups of the PEG, at 1.5 ppm, the peaks of the methyl groups of lactic acid were observed. At 4.8 ppm, the peaks corresponded to –CH of glycol acid, and at 5.2 ppm to –CH of lactic acid (Figure 2A). The ^19^F-NMR spectrum of PLGA-PEG-TFA showed the typical peak of –CF_3_ from TFA at −74.4 and −74.5 ppm (Figure 2B). TFA was chemically incorporated onto the copolymer. Therefore, we confirm that a new intrinsically labeled copolymer was obtained, thereby allowing for the visualization of NPs using molecular imaging and ^19^F-MRI.

### 3.2. Preparation and Characterization of NPs

In this study, PLGA-PEG NPs and novel PLGA-PEG-TFA NPs were designed to be detected by molecular imaging techniques in vitro and in vivo. The two types of NP were prepared using a single emulsion solvent evaporation method. The method used to prepare the NPs allowed the formation of hydrophobic chains of PLGA in the core of the NPs and a hydrophilic chain of PEG-TFA on the surface. The use of PEG chains on NPs leads to a hydrated cloud with a large volume that sterically precludes interaction with closer NPs and blood components [19]. Moreover, the use of PEGylation increases the blood circulation time, for systemic and local administration of NPs [13,20].

The average diameter of PLGA-PEG NPs (control) was 170 ± 5 nm and PLGA-PEG-TFA NPs (formulated with novel polymer) 203 ± 2 nm; their polydispersity was 0.4 and 0.2 respectively, suggesting uniformity of the particle size distribution [21]. The diameter of the two types of NP is in the range of nanomaterial suitable for cellular uptake via endocytosis [22]. All the NPs carried a negative charge. The Zeta potential for PLGA-PEG NPs was −16.9 ± 3.6 mV while for PLGA-PEG-TFA NPs it was −21 ± 1.8 mV, indicating that the NPs have good stability in water solutions [23,24,25,26]. The encapsulation efficiency (EE%) for fluorescent NIR dye was 57.6% for PLGA-PEG NPs and 44.4% for PLGA-PEG-TFA NPs, thereby, indicating efficiency encapsulation (Table 1).

To confirm the presence of TFA on the PLGA-PEG copolymer a relaxivity measurement was performed, indicated that T1 and T2-weighted MR measurements were for solo TFA T1 = 854.500 ms; T2 = 579.699 ms, detect at −74.092 and, for PLGA-PEG-TFA NPs T1 = 1.135 s; T2 = 886.700 ms, detected at −74.284 ppm. The morphology of the NPs was confirmed by transmission electron microscopy (TEM). TEM imaging revealed the uniform size and spherical shape of PLGA-PEG-TFA NPs (Figure 3A). The morphology of the NPs was also confirmed using atomic force microscopy (AFM) (Figure 3B). The size of PLGA-PEG-TFA NPs, as determined by TEM and AFM, was slightly smaller than that determined by DLS. This difference is attributed to different preparation methods. In contrast to TEM and AFM, measurement by DLS required NPs to be dissolved in MilliQ water (Figure 3 and Table 1).

The study of NIR dye release from PLGA-PEG and PLGA-PEG-TFA NPs in PBS showed different patterns (Figure 3D). Significant differences were observed between the two types of NP in the first phase of the release at days 2, 4, and 7. The PLGA-PEG NPs showed a faster release of NIR dye compared to PLGA-PEG NPs. After 4 days, PLGA-PEG NPs had released almost 40% of the NIR dye and PLGA-PEG-TFA NPs almost 15%. This difference in release could be explained by polymer composition. In this regard, the surface of PLGA-PEG-TFA NPs presents TFA, which can osculate and slow down NP release. However, after 10 days, there were no significant differences observed between the two types of NP, and after 30 days both types had released the same amount of NIR dye (around 75%). The difference in release behavior between PLGA-PEG-TFA NPs and PLGA-PEG NPs could be caused by drug diffusion, polymer composition and degradation.

### 3.3. Characterization of Novel Polymeric NPs in the In Vitro Experiments

#### 3.3.1. Cellular Metabolic Assay

The MTS assay showed that all the NPs were non-cytotoxic to the cells at the studied time points. In the first 24 and 48 h, all the concentrations of PLGA-PEG NPs and PLGA-PEG-TFA NPs tested exhibited a non-significant reduction in cell metabolic activity, followed by a slow increase with time, which was probably due to proliferation (Figure 4A,B). PLGA-PEG are copolymers present in several FDA-approved drug formulations, and in recent years many studies have shown that they are highly biocompatible and suitable for in vitro and in vivo treatment. In an aqueous environment, PLGA is degraded into lactic acid and glycolic acid. These two compounds enter the tricarboxylic acid cycle of the chondrocyte and are eliminated as carbon dioxide and water [11]. This can explain the increase in the metabolic activity of C28/I2 cells after 72 h of incubation with NPs (Figure 4C).

#### 3.3.2. NP Uptake by Chondrocytes

To evaluate NP uptake, C28/I2 cells were incubated with the NPs and their fluorescence intensity was measured at four time points (Figure 5A). Cell uptake of PLGA-PEG and PLGA-PEG-TFA NPs increased over time, except for PLGA-PEG NPs at the time point of 24 h. No significant differences were observed between the two types of NP after 1 h, 4 h, or 8 h of incubation. However, after 24, 48 h, and 72 h PLGA-PEG-TFA NPs showed significantly higher fluorescence than PLGA-PEG NPs.

To confirm the results obtained by fluorescence, a confocal and fluorescence microscopy characterization was performed (Figure 5B,C). After 24 h of incubation, cells were observed to take up NPs. Both types of NP were present in the cell membrane. Interesting, and same results were observed for all the uptake assays. The PLGA-PEG-TFA NPs showed high cellular uptake after 24 h of incubation (Figure 5A,B2,C2). This phenomenon is probably due to the higher negative surface charge of these NPs (Table 1) [27]. Taken together, the data indicate that the PLGA-PEG-TFA NPs induced the slow release of the encapsulated compounds, higher cell metabolic activity, and higher cellular uptake compared to PLGA-PEG NPs.

### 3.4. In Vivo and Ex Vivo Imaging of PLGA-PEG NPs Injected into a Mouse Model of Osteoarthritis

After confirming that DMM successfully induced knee OA joint in mice, the legs were scanned with µCT (Figure 6A). It is known that OA induces significant changes to bone structure [28]. The μCT-scan was used to visualize the bone structure of the knee joint (Figure 6A). The results show that DMM induced knee OA since osteophyte formation and cartilage degeneration were observed in the joints, while these phenomena were not observed in healthy knees (Figure 6A1,A2).

Afterwards, the animals were treated with the two types of NPs and monitored for 168 h. The mice received IA injections of the NPs. The new intrinsically labeled polymeric NPs (PLGA-PEG-TFA) and the PLGA-PEG NPs were strongly detected in live mice using a Pearl Imaging System at 800 nm (Figure 6A). The mice were divided into distinct groups (*n = 3*) and scanned. They were then sacrificed after 2, 48, and 166 h. No signal was detected for control mice (Figure 6B1). A gradual decline of fluorescent signal was observed over time, but it was still highly detectable in vivo at the latest time point, especially for the knee injected with PLGA-PEG-TFA NPs (Figure 6B2,B3). Those results are indicating that the NP are releasing the NIR dye. After mouse sacrifice, the knee joint and the main organs were collected and scanned at 800 nm using an Odyssey Imaging System (Figure 6C,D).

To monitor the polymeric NPs and detect their localization in the knee joint, the legs were scanned with ^19^F-MRI (Figure 7A) and then subjected to histological analysis (Figure 7B). The ^19^F-MRI image acquisition was done immediately after the mice had been sacrificed. Figure 7A shows high-resolution images of the legs. The images were acquired at 2, 48 and 168 h after IA injection of PLGA-PEG-TFA NPs in the right knee. As expected, it is not possible to distinguish the specific position of the NPs in the knee joint from either in vivo or ex vivo images. However, it was observed that the PLGA-PEG-TFA NPs remained in the knee joint for more than 166 h. The results indicate that the ^19^F signal (in red) decreased gradually, suggesting that the PLGA-PEG-TFA NPs were successfully visualized. Compared to the PLGA-PEG NPs, the multi-modal PLGA-PEG-TFA NPs can be detected by optical imaging and by ^19^F-MRI. Both are non-invasive molecular imaging techniques. The ^19^F-MRI molecular imaging technique seems to have better advantages in the use compared to optical imaging. The main advantages are: ^19^F-MRI produces high spatial resolution images in three orientations: Axial, coronal, and sagittal. This technique does not use radiation and provides soft tissue contrast as compared to µCT, has unlimited tissue penetration and it can determine NPs biodistribution at the tissue level. (Information about this topic can be found elsewhere, e.g., [28,29]). In Figure 7B1, hematoxylin/Eosin and F4/80 stained slides to visualize macrophages, and in Figure 7B2, Safranin-O staining was used to visualize the cartilage and osteophytes. These results are in line with the observation obtained with in-vitro studies: (1) The healthy knee did not show the presence of macrophages and osteophytes, (2) the OA knee with no treatment showed a high concentration of macrophages, cartilage degeneration, and osteophytes, (3) the OA knee treated with NPs showed a minor concentration of macrophages, suggesting that PLGA-PEG-TFA NPs are not toxic to cells in-vivo. From the Safranin-O staining (PLGA-PEG-TFA NPs injected) fewer osteophytes formation was observed and cartilage damage is present compared to the OA knee with no treatment. This result is due to the short time of this experiment. Nevertheless, these results are showing that injection of the PLGA-PEG-TFA NPs did not worsen the progression of OA in mice (Figure 7B). 

## 4. Conclusions

We designed and developed a new intrinsic co-polymer, namely PLGA-PEG-TFA, that allowed us to trace the NPs by ^19^F-MRIIt was successfully visualized and traced in vitro, in vivo, and ex vivo by optical imaging and ^19^F-MRI in an osteoarthritic knee joint mice model. The NPs were not toxic to chondrocytes in vitro. Moreover, PLGA-PEG and PLGA-PEG-TFA NPs were loaded with NIR dye, which allowed the measurement of their biodistribution and retention over time in the knee joint, as well as their visualization by ^19^F-MRI. In this study, it was not determined which cell populations or extracellular structures would account for the differential uptake of the NPs, which is very interesting and should be determined in a follow-up study. Taken together, this study presents a proof-of-principle concept of a NP mediated multi-modal modality for the visualization of OA and therapy that could complement or challenge existing multimodal imaging approaches (optical imaging, MRI, µ-CT).

## Figures and Tables

**Figure 1 pharmaceutics-13-00235-f001:**
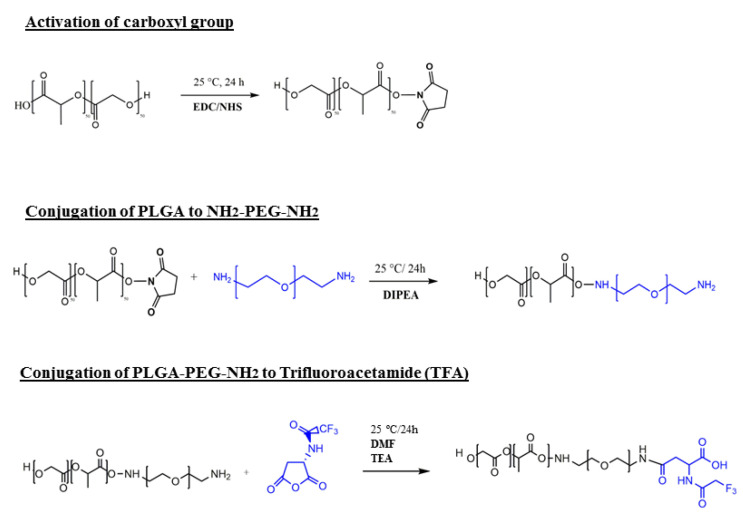
Schematic representation and synthetic procedure of PLGA-PEG-TFA (poly (lactic-co-glycolic acid)-polyethylene glycol-trifluoroacetamide) formulation.

**Figure 2 pharmaceutics-13-00235-f002:**
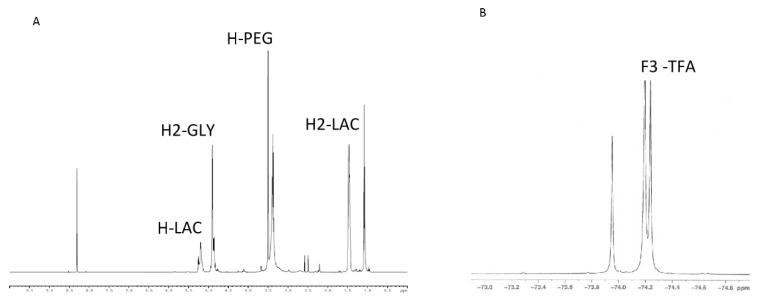
NMR characterization of functional copolymers. (**A**) ^1^H-NMR spectrum of PLGA-PEG-TFA in DMSO. (**B**) ^19^F-NMR spectrum of PLGA-PEG-TFA in D_2_O.

**Figure 3 pharmaceutics-13-00235-f003:**
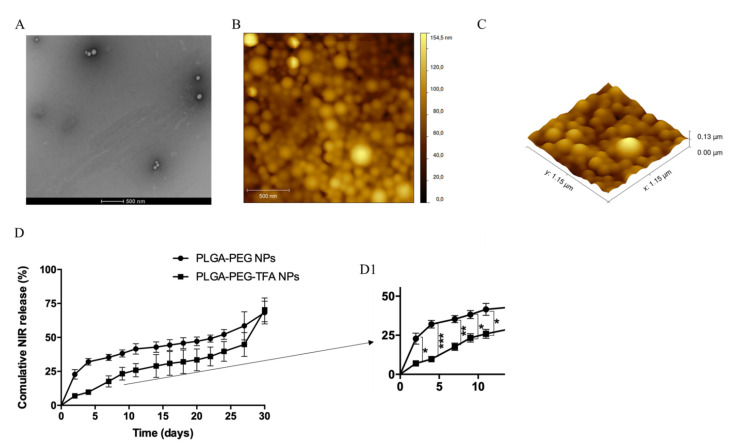
Characterization of PLGA-PEG-TFA NPs with (**A**) images of nanoparticle (NP) morphology obtained by TEM; (**B**) AFM 2D image; and (**C**) AFM 3D image of functional PLGA-PEG-TFA NPs. (**D**) Cumulative release study of NIR dye in PBS solution from PLGA-PEG and PLGA-PEG-TFA NPs. (**D1**) Zoom image of the cumulative release study of NIR dye with the significant differences. Asterisks indicate significant difference in Student’s *t*-test, * *p* < 0.05, ** *p* < 0.01, *** *p* < 0.001.

**Figure 4 pharmaceutics-13-00235-f004:**
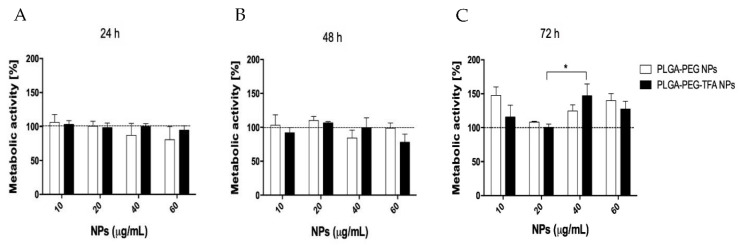
Cell metabolic activity assay (MTS) of the PLGA-PEG NPs and PLGA-PEG-TFA NPs loaded with NIR dye at different concentrations incubated with C28/I2 cell line. The percent of metabolic cell activity was measured at three time point. (**A**) 24 h, (**B**) 48 h, and (**C**) 72 h. For each assay, the positive control was obtained using DMSO 50% and the negative control was obtained with cells only. The negative controls are indicated by the black horizontal line. Asterisk indicates significant difference in Student’s *t*-test, * *p* < 0.05.

**Figure 5 pharmaceutics-13-00235-f005:**
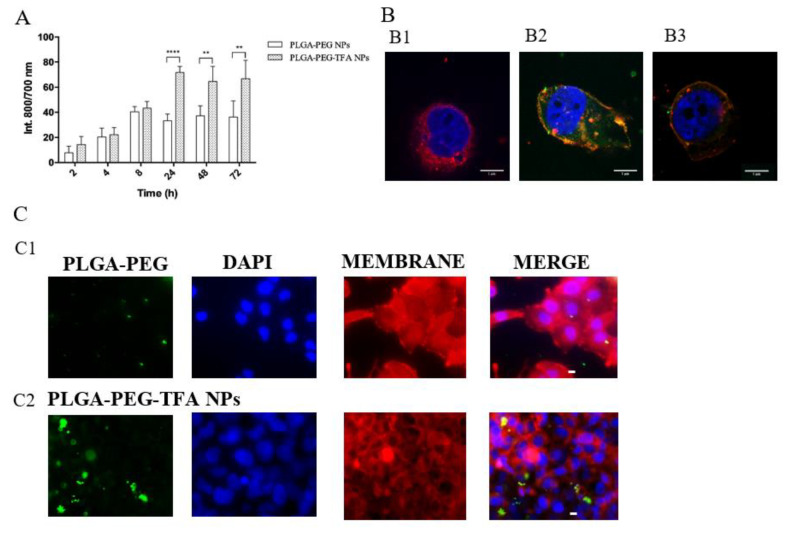
NP uptake assays. (**A**) PLGA-PEG NP and PLGA-PEG-TFA NP uptake assay characterized using an Odyssey Imaging System. Both types of NP were incubated with C28/I2 cells for 2, 4, 8, 24, 48, or 72 h. (**B**) Confocal microscopy images of cell membrane stained with DiD (red) (644 nm excitation, emission filter 670/40), nucleus stained with DAPI (blue) (365 nm excitation, 40 ms, emission filter 450/50) and FITC NPs (green) (490 nm excitation, emission filter 515). Images taken after 24 h of cell incubation with NPs. The images were taken with a Leica SP5C Spectral Confocal Laser Scanning Microscope. (**B1**) Control sample (only cells with no NP treatment), (**B2**) cells and PLGA-PEG-TFA NPs, and (**B3**) cells and PLGA-PEG NPs. (**C**) Fluorescence microscopy images showing internalization of fluorescent PLGA-PEG and PLGA-PEG-TFA NPs. Visualization after 24 h of incubation with NPs in PBS. (**C1**) PLGA-PEG NPs, and (**C2**) PLGA-PEG-TFA NPs. Cell membrane stained with CD44-PE (red), cell nucleus with DAPI (blue) and NPs with NIR dye (green). The images were taken with a Leica fluorescence microscope at 43× magnification. Asterisks indicate significant difference in two-way ANOVA, ** *p* < 0.01 and **** *p* < 0.0001.

**Figure 6 pharmaceutics-13-00235-f006:**
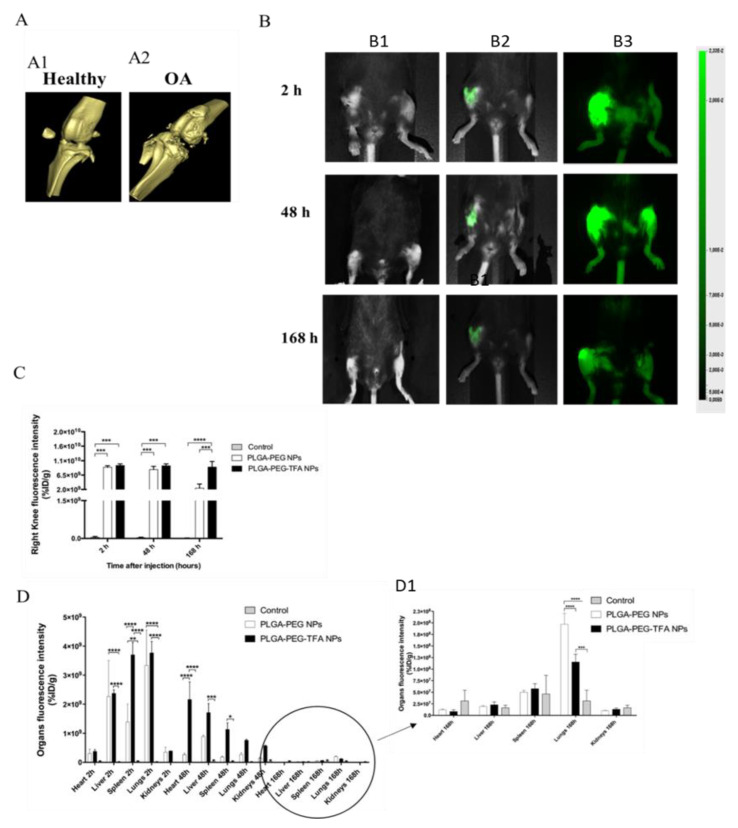
Representative mouse knee images in Ex-vivo and In-vivo. (**A**) Ex vivo µCT- scan of knee joints showing a 3D view of the cartilage surface. (**A1**) scan of healthy knee joint (negative control group) and (**A2**) scan of OA knee joint after 3 weeks of destabilization of medial meniscus (DMM) induction (positive control group). (**B**) In vivo images using the Pearl Image System were acquired at 800 nm. Mice received intra-articular (IA) injection of these NPs after DMM induction. From top to bottom, 800 nm scan of mice 2, 48, and 168 h after injection with PLGA-PEG NPs and PLGA-PEG-TFA NPs (in green). (**B1**) control group, (**B2**) PLGA-PEG NPs group and (**B3**) PLGA-PEG-TFA NPs group. (**C**) Quantification of fluorescent NP release in the knee joints at three time points. (**D**) Biodistribution of fluorescent NPs in main organs at three time points. (**D1**) Zoom image of biodistribution of fluorescent NPs in main organs 168 h post-injection. Asterisks indicate significant difference in two-way ANOVA, * *p* < 0.05, ** *p* < 0.01, *** *p* < 0.001 and **** *p* < 0.0001.

**Figure 7 pharmaceutics-13-00235-f007:**
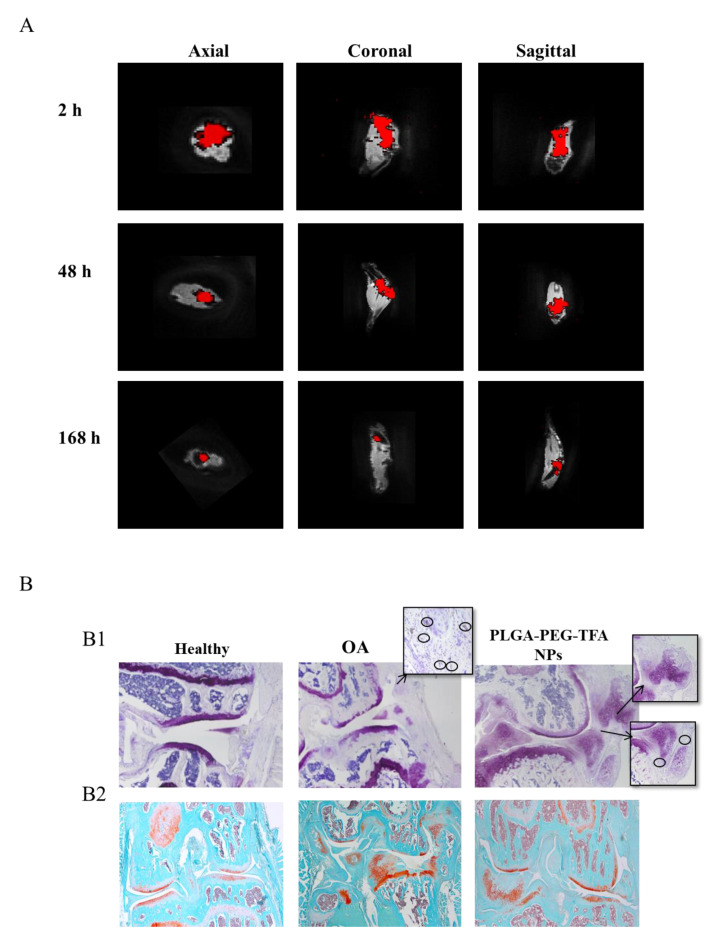
Representative images of PLGA-PEG-TFA NPs in ex vivo. (**A**) Ex vivo ^19^F-MRI images in axial, coronal and sagittal view of the right knee. The image was taken at 2, 48, and 168 h post-NP injection. Red shows the signal of TFA present on the surface of PLGA-PEG-TFA NPs. (**B**) Histological staining images. From left to right healthy knee (negative control), osteoarthritis (OA) knee without NPs treatment (positive control), representative pictures of OA knee treated with PLGA-PEG-TFA NPs. (**B1**) Hematoxylin/Eosin and F4/80 staining, macrophages are indicated with the black circle. (**B2**) Histological staining with Safranin-O fast green.

**Table 1 pharmaceutics-13-00235-t001:** Characteristics of the PLGA-PEG and PLGA-PEG-TFA NPs.

NPs	Particle Size (nm)	PDI	ζ Potential (mV)	EE% (NIR)	EE% (FITC)
PLGA-PEG	169.7 ± 5.4	0.30 ± 0.01	−16.9 ± 3.6	57.6	26.3
PLGA-PEG-TFA	202.8 ± 1.7	0.20 ± 0.03	−21.3 ± 1.8	44.4	12.5

## Data Availability

Data sharing not applicable.

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
