# Peer review of "Novel Fluorinated Poly (Lactic-Co-Glycolic acid) (PLGA) and Polyethylene Glycol (PEG) Nanoparticles for Monitoring and Imaging in Osteoarthritis"

_pharmaceutics, 2021, doi:10.3390/pharmaceutics13020235_

Round 1
Reviewer 1 Report
This is a really clear presentation of the synthesis and delivery of a contrast agent for Fluorine MRI imaging. I can not comment on the chemistry, but it appears logical based on the success of other PGLA formulations in soft and hard tissue drug delivery.
The cell based experiments clearly show the uptake of contrast material into "naked" chondrocytes in cell culture; however, this is a very crude evaluation as chondrocyte membranes have very limited exposure to the synovial fluid where contrast material would be injected. Dense collagen fibres and extracellular matrix abound, making access to chondrocytes difficult. This could be an advantage in ultra-high resolution imaging where intact superficial cells might take up contrast material in normal cartilage, but damaged cartilage would demonstrate an entirely different pattern. I am disappointed that the authors did not conduct ex vivo studies in explants that would have clarified this situation--Please consider this as these experiments would solidify the authors conceptual approach to targeted imaging in OA.
The in vivo experiments demonstrate a depot effect of the contrast agent that persists for a relatively long time. The resolution of the imaging is insufficient to determine what tissues harbour the contrast material. Post mortem dissection and imaging of isolated synovial membrane, cartilage, lymph nodes, joint capsule etc. would elucidate the depot effect. It is likely that the depot effect here is not the cartilage but the phagocytic cells in the synovial membrane.
There is an unexplained cross over to the contralateral knee joint that needs to be explains and is a potentially confounding effect if the contrast material is being redistributed to other joints.
Author Response
Reviewer #1 This is a really clear presentation of the synthesis and delivery of a contrast agent for Fluorine MRI imaging. I cannot comment on the chemistry, but it appears logical based on the success of other PGLA formulations in soft and hard tissue drug delivery.
Authors: Dear Reviewer #1, we are very appreciated for your careful review of the paper. We took your comments, critics and suggestion into consideration and we think that they helped to improve the manuscript considerably. We reply to your detailed comments below.
- The cell-based experiments clearly show the uptake of contrast material into "naked" chondrocytes in cell culture; however, this is a very crude evaluation as chondrocyte membranes have very limited exposure to the synovial fluid where contrast material would be injected. Dense collagen fibres and extracellular matrix abound, making access to chondrocytes difficult. This could be an advantage in ultra-high-resolution imaging where intact superficial cells might take up contrast material in normal cartilage, but damaged cartilage would demonstrate an entirely different pattern. I am disappointed that the authors did not conduct ex vivo studies in explants that would have clarified this situation--Please consider this as these experiments would solidify the authors conceptual approach to targeted imaging in OA.
Authors: We thankful the reviewer for the observation suggested, we performed the histological study. The Hematoxylin/ Eosin and F4/80 staining was performed for the investigation of macrophages formation and, the Safranin-O staining to visualize the cartilage and osteophytes in mice knee joint. Please, see section Material and Methods paragraph 2.6.4, Result and discussion paragraph 3.4 and Figure 7B.
- The in vivo experiments demonstrate a depot effect of the contrast agent that persists for a relatively long time. The resolution of the imaging is insufficient to determine what tissues harbour the contrast material. Postmortem dissection and imaging of isolated synovial membrane, cartilage, lymph nodes, joint capsule etc. would elucidate the depot effect. It is likely that the depot effect here is not the cartilage but the phagocytic cells in the synovial membrane. There is an unexplained cross over to the contralateral knee joint that needs to be explains and is a potentially confounding effect if the contrast material is being redistributed to other joints.
Authors: We thankful the reviewer for the observation suggested, we performed the histological study as mention in the previous answer.
Reviewer 2 Report
The work titled as “Novel fluorinated poly (lactic-co-glycolic acid) (PLGA) and polyethylene glycol (PEG) nanoparticles for monitoring and imaging in osteoarthritis”, described the synthesis of novel fluorinated polymeric NPs, and their applications as vectors for multi-modal nanoprobes. The involved polymeric NPs were modified with trifluoroacetamide (TFA), with loaded/coupled dye for different imaging modalities, such as magnetic resonance imaging (MRI) and optical imaging. The work can be published after clarification the following items.
- In Figure 6, although NPs loaded the NIR dye can monitor osteoarthritis, but there is no obviously differentiation between PLGA-PEG NPs and PLGA-PEG-TFA NPs. So what is the advantage for the fluorinated PLGA-PEG NPs (PLGA-PEG-TFA NPs)?
- The author claimed that the TFA was present on the shell of NPs. Are there any evidence or based in assumption?
- Could the authors put the control images, PLGA-PEG NPs and PLGA-PEG-TFA NPs in Figure 6A?
Author Response
Reviewer #2, The work titled as “Novel fluorinated poly (lactic-co-glycolic acid) (PLGA) and polyethylene glycol (PEG) nanoparticles for monitoring and imaging in osteoarthritis”, described the synthesis of novel fluorinated polymeric NPs, and their applications as vectors for multi-modal nanoprobes. The involved polymeric NPs were modified with trifluoroacetamide (TFA), with loaded/coupled dye for different imaging modalities, such as magnetic resonance imaging (MRI) and optical imaging. The work can be published after clarification the following items.
Authors: Dear Reviewer #2, we are very appreciated for your careful review of the paper. We
took your comments, critics and suggestion into consideration and we think that they helped to
improve the manuscript considerably. We reply to your detailed comments below.
- In Figure 6, although NPs loaded the NIR dye can monitor osteoarthritis, but there is not obviously differentiation between PLGA-PEG NPs and PLGA-PEG-TFA NPs. So, what is the advantage for the fluorinated PLGA-PEG NPs (PLGA-PEG-TFA NPs)?
Authors: Thank you for your comment. We decided to investigate both PLGA-PEG and PLGA-PEG-TFA nanoparticles because we wanted to compare them in-vitro and in-vivo. The PLGA-PEG NPs can only be detected by Optical imaging and the PLGA-PEG-TFA NPs can be detected with two molecular imaging techniques: Optical imaging and 19F-MRI.
The main advantages to using 19F-MRI instead the optical imaging is:
- Produces high spatial resolution images compared to other techniques such as optical or radionuclide imaging
- Provides better soft tissue contrast than CT and can differentiate better between fat, water, muscle, and soft tissue
- Not limited by tissue depth (unlimited penetration)
- Can determine nanoparticle biodistribution at the tissue or organ level
- Biodistribution of nanoparticles can be assessed in real-time and over multiple time points
Information about this topic can be found elsewhere, e.g Arms L, Smith DW, Flynn J, et al. Advantages and Limitations of Current Techniques for Analyzing the Biodistribution of Nanoparticles. Front Pharmacol. 2018;9:802. Published 2018 Aug 14. doi:10.3389/fphar.2018.00802.
The authors included this information in the main text. Please, see section Results and discussion, paragraph 3.4.
- The author claimed that the TFA was present on the shell of NPs. Are there any evidence or based in assumption?
Authors: Thank you for your comment. We have strong evidences that the TFA is bound to the C-terminal of the PLGA-PEG co-polymer due to the Relaxivity study that we performed on the formulated nanoparticles. Please, see section Material and Methods 2.4.5 and section Result and discussion paragraph 3.2. Therefore, we are completely sure that the TFA is on the NPs surface i) due to the Relaxivity study and ii) for the formulation method of our NPs.
- Could the authors put the control images, PLGA-PEG NPs and PLGA-PEG-TFA NPs in Figure 6A?
Authors: To accommodate the reviewer’s request we included the control images in the figure, and we adjusted the text, accordingly. Please, see Figure 6B.

Reviewer 3 Report
This is an interesting article revealing the development and evaluation of fluorinated PLGA and PEG NP as a monitoring tool for joint osteoarthritis. However, there are some controversial parts that needed to be clarified.
Firstly, there is an absence of discussion part?! I took this as a major mistake and believe that the authors did not pay full responsibility for checking their manuscript before submission.
For the result portion:
- Problem of MTS assay as shown in Figure 4: there was a missing of control group, which is the non-treated cells. The viability of cells treated with either NPs should be compared to non-treated control.
- NP uptake by chondrocytes, the data only show the NP uptake up to 24 hours. Longer study period may be needed.
- there was an inconsistency of methodology. in the MM part, the authors stated that the mice received IA injection of NPs 3 weeks after DMM, but in the result part, it becomes 2 weeks? ???
- Any evidence show that 2 or 3 weeks of DMM sufficient to induce OA?
- for the in vivo section, the results only showed that the IA injection of NPs could be visualized by the imaging system but it does not equivalent that OA could be visualized through NPs injection! otherwise, the control knees (non DMM knee) should be IA injected and checked ! please show the data if available or the whole hypothesis of this manuscript could not be established at all!
- the MRI images in Figure 7 were not readable or interpretable. Please provide clear images with better magnification and resolution.
Author Response
Reviewer #3 This is an interesting article revealing the development and evaluation of fluorinated PLGA and PEG NP as a monitoring tool for joint osteoarthritis. However, there are some controversial parts that needed to be clarified.
Firstly, there is an absence of discussion part?! I took this as a major mistake and believe that the authors did not pay full responsibility for checking their manuscript before submission.
Dear Reviewer#3, we are very appreciated for your careful review of the paper. We took your comments, critics and suggestion into consideration and we think that they helped to improve the manuscript considerably. We reply to your detailed comments below.
For the result portion:
- Problem of MTS assay as shown in Figure 4: there was a missing of control group, which is the non-treated cells. The viability of cells treated with either NPs should be compared to non-treated control.
Authors: The non-treated cells group, were investigated for the MTS assay and compared with the cell NPs treated. Please, see section Material and Methods paragraph. 2.5.2. Cell metabolic assay (MTS) and, Figure 4A, 4B and 4C the negative controls (non-treated cells) are indicated by the black horizontal lines.
- NP uptake by chondrocytes, the data only show the NP uptake up to 24 hours. Longer study period may be needed.
Authors: Thank you for your comment. The authors included a longer NPs uptake studies characterized by optical imaging. The study was performed for longer time, adding 48 h and 72 h, same to the in-vivo study. Please, see section Materials and Methods paragraph 2.5.3, section Results and discussion paragraph 3.3.2 and Figure 5A.
- there was an inconsistency of methodology. in the MM part, the authors stated that the mice received IA injection of NPs 3 weeks after DMM, but in the result part, it becomes 2 weeks????
Authors: Apologies for the confusion, we adjust the text. Please, see section Results and Discussions paragraph 3.4. In vivo and ex vivo imaging of PLGA-PEG NPs injected into a mouse model of osteoarthritis. According to published studies, osteoarthritis is inducted after 3 weeks of DMM and we followed their studies. Moreover, In figure 6A, a µCT- scan for positive and negative control mouse knee. The positive control, µCT- scan of osteoarthritic knee joint, is clearly showing the cartilage defects characterized by OA after 3 weeks of DMM induction.
- Any evidence shows that 2 or 3 weeks of DMM sufficient to induce OA?
Authors: Thank you for your comment. Yes, published studies suggested that non-sever osteoarthritis can be inducted after 3 weeks from the DMM surgery in mice knee joint.
Information can be found in the following papers referred:
- Fang, H., Huang, L., Welch, I. et al. Early Changes of Articular Cartilage and Subchondral Bone in The DMM Mouse Model of Osteoarthritis. Sci Rep 8, 2855 (2018). https://doi.org/10.1038/s41598-018-21184-5
- Iijima H, Aoyama T, Ito A, Tajino J, Nagai M, Zhang X, Yamaguchi S, Akiyama H, Kuroki H. Destabilization of the medial meniscus leads to subchondral bone defects and site-specific cartilage degeneration in an experimental rat model. Osteoarthritis Cartilage. 2014 Jul;22(7):1036-43. doi: 10.1016/j.joca.2014.05.009. Epub 2014 May 21. PMID: 24857975.
Moreover, apologies for the confusion in the previous text. The µCT-scan was performed to confirm that after 3 weeks from the DMM the mice knee had developed osteoarthritis. Please, see section Result and discussion paragraph 3.4 and Figure 6.
- for the in vivo section, the results only showed that the IA injection of NPs could be visualized by the imaging system but it does not equivalent that OA could be visualized through NPs injection! otherwise, the control knees (non DMM knee) should be IA injected and checked ! please show the data if available or the whole hypothesis of this manuscript could not be established at all!
Authors: We appreciate the suggestion made by the reviewer. A control image (healthy mouse with non- nanoparticles injection), the PLGA-PEG NPs and PLGA-PEG-TFA NPs group injected were added in the figure. Please, see figure 6B and, the text was adjusted accordingly, see paragraph 3.4. In our previous published study, we did a more detailed mice study please, see the referred paper:
Zerrillo L, Que I, Vepris O, Morgado LN, Chan A, Bierau K, Li Y, Galli F, Bos E, Censi R, Di Martino P, van Osch GJVM, Cruz LJ. pH-responsive poly(lactide-co-glycolide) nanoparticles containing near-infrared dye for visualization and hyaluronic acid for treatment of osteoarthritis. J Control Release. 2019 Sep 10;309:265-276. doi: 10.1016/j.jconrel.2019.07.031. Epub 2019 Jul 27. PMID: 31362078.
Moreover, the goal of this paper was to prove that PLGA-PEG-TFA NPs can be detected in osteoarthritic knee by optical imaging and F-MRI. Because of this and our previous published study we did not injected healthy mice with NPs.
- the MRI images in Figure 7 were not readable or interpretable. Please provide clear images with better magnification and resolution.
Authors: Thank you for your comment. Unfortunately, we don’t have a better magnification and resolution images for the 19F-MRI. However, this is a typical and maximal magnification and resolution that we can get with the current 19F coin.
Round 2
Reviewer 1 Report
The revised manuscript provides additional rationale and explanation of the findings. There is little evidence to indicate that the method is actually imaging cartilage, so absolutely no reference or statement to that effect should be included in the manuscript. Its very likely the imaging method provides tracking of immune cell mediated inflammation and synovitis, so until the authors show that their data differs from an inflammatory model such as carrgeenan or urate crystals or collagen readers should harbour skepticism about the specificity of the these nanoparticles. I think the authors need to make a clear statement in the conclusion about what they think they are imaging and what further studies would be needed to confirm that OA is the target.
Author Response
Dear Reviewer,
We hope that this time we addressed correctly your comment.
Please, see the attachment file and the manuscript.
Thank you in advance
Best regards
Attachement:
Authors: Dear Reviewer #1, we are very appreciated for your careful review of the paper. We took your comments, critics and suggestion into consideration and we think that they helped to improve the manuscript considerably. We reply to your comment below.
Reviewer #1 The revised manuscript provides additional rationale and explanation of the findings. There is little evidence to indicate that the method is actually imaging cartilage, so absolutely no reference or statement to that effect should be included in the manuscript. It’s very likely the imaging method provides tracking of immune cell mediated inflammation and synovitis, so until the authors show that their data differs from an inflammatory model such as carrageenan or urate crystals or collagen readers should harbour skepticism about the specificity of the these nanoparticles. I think the authors need to make a clear statement in the conclusion about what they think they are imaging and what further studies would be needed to confirm that OA is the target.
Authors: We thankful the reviewer for the observation suggested, we hope this time to address your comment. Please, see the conclusion section.
“In this study, it was not determined which cell populations or extracellular structures would account for the differential uptake of the NPs, which is very interesting and should be determined in a follow-up study. Taken together, this study presents a proof-of-principle concept of a NP mediated multi-modal modality for the visualization of OA and therapy that could complement or challenges existing multimodal imaging approaches (optical imaging, MRI, µ-CT)”.
Reviewer 3 Report
All revisions had been done according to reviewer's suggestions
Author Response
Dear Reviewer,
The paper was reviewed by an English writer.
Please, see the manuscript.
Thank you in advance.